# First-Principles Study on the Photoelectric Properties of CsGeI₃ under Hydrostatic Pressure

**Li-Ke Gao [1], Yan-Lin Tang [2,*] and Xin-Feng Diao [1]**

[1] College of Big Data and Information Engineering, Guizhou University, Guiyang 550025, China; 20150685@git.edu.cn (L.-K.G.); diaoxinfeng@gznc.edu.cn (X.-F.D.)

[2] College of Physics, Guizhou University, Guiyang 550025, China

[*] Correspondence: yltang@gzu.edu.cn

**Abstract:** CsGeI₃ has been widely studied as an important photoelectric material. Based on the density functional theory (DFT), we use first-principles to study the photoelectric properties of CsGeI₃ by applying successive hydrostatic pressure. It has been found that CsGeI₃ has an optimal optical band gap value of 1.37 eV when the applied pressure is −0.5 GPa, so this paper focuses on the comparative study of the photoelectric properties when the pressure is −0.5 GPa and 0 GPa. The results showed that CsGeI₃ has a higher dielectric value, conductivity, and absorption coefficient and blue shift in absorption spectrum when the pressure is −0.5 GPa. By calculating and comparing the effective masses of electrons and holes and the exciton binding energy, it was found that their values are relatively small, which indicates that CsGeI₃ is an efficient light absorbing material. CsGeI₃ was found to be stable under both pressure conditions through multiple calculations of the Born Huang stability criterion, tolerance factor T, and phonon spectrum with or without virtual frequency. We also calculated the elastic modulus of both pressure conditions and found that they are both soft, ductile, and anisotropic. Finally, the thermal properties of CsGeI₃ under two kinds of pressure were studied. It was found that the Debye temperature and heat capacity of CsGeI₃ increased with the increase of thermodynamic temperature, and the Debye temperature increased rapidly after pressure, while the heat capacity slowly increased and finally stabilized. Through the calculation of enthalpy, entropy, and Gibbs free energy of CsGeI₃, it was found that the Gibbs free energy decreases faster with the increase of temperature without applied pressure, which indicates that CsGeI₃ has a higher stability without pressure. Through the comparative analysis of the photoelectric properties of CsGeI₃ under pressure, it was found that CsGeI₃ after applied pressure is a good photoelectric material and suitable for perovskite solar cells (PSCs) material.

**Keywords:** DFT; CsGeI₃ perovskite; hydrostatic pressure; photoelectric properties

## 1. Introduction

Organic–inorganic hybrid halide perovskites and related materials have attracted much attention in the past decade due to their high absorption coefficient, appropriate band gap width, excellent carrier mobility, long carrier life, and low cost [1–6]. Recent studies have shown that they have a high photoelectric conversion efficiency (PCE) of more than 25% [7]. They have been widely used in solar cells, photon emitters, photodetectors, and other photoelectric devices [8–11]. In general, the most used perovskite materials are organic methylammonium lead iodide and formamidinium lead iodide due to their excellent PCE. However, the organic cations methylammonium cation (MA⁺) and formamidinium cation (FA⁺) are easily decomposed due to sun and rain exposure after prolonged exposure to the air. At present, inorganic perovskite has attracted research attention, among which Cs⁺ is the most studied substitute for MA⁺ and FA⁺. In fact, C$_S^+$ (1.67 Å) has a smaller ionic radius than

$MA^+$ (1.80 Å) or $FA^+$ (1.90 Å), and the tolerance factor of $CsGeI_3$ is 0.81 calculated by the Goldschmidt formular [12,13]. Due to its high thermal stability, researchers have studied the inorganic perovskite $CsPbI_3$ [14–17], in which the PCE of quantum dot devices [16] is as high as 13.4%, and that of thin films [15] is as high as 15.1%. The search for non-toxic PSCs materials has become a research hotspot because $CsPbI_3$ contains the toxic element Pb. We know that Ge, Sn, and Pb are in the same main group, and that they have similar properties. For example, they have similar electron shells and their outermost shell is composed of two electrons. As an efficient hole transport material, $CsGeI_3$ can be applied to solar cells due to its small effective hole mass, without low-energy deep hole trap and the suitable band offset with solar absorber materials such as dye molecules and methylammonium lead iodide [18]. Krishnamoorthy et al. synthesized $CsGeI_3$ in their experiment and studied its PCE. They found that the PCE reached 11% and showed good stability [19].

Most researchers improve PCE by doping materials. However, the application of pressure can change the structure of PSCs, thus changing their photoelectric properties. Many experiments have been carried out to apply pressure to methylammonium lead iodide to change its electronic structure and photoelectric properties [20,21]. Schwarz and Seo studied the geometry and band structures of $CsGeBr_3$ and $CsGeCl_3$ under pressure [22,23]. Jing et al. studied the photoelectric properties of $CsPbI_3$ at different pressure by the first-principles and found that when the pressure was 1.4 GPa, it had an optimal band gap value of 1.34 eV [24].

However, there are few studies on the photoelectric properties of $CsGeI_3$ under pressure. Although Liu et al. studied some photoelectric properties of $CsGeI_3$ under hydrostatic pressure through first-principles, we still do not know enough about it [25]. Based on the first-principles, the change of photoelectric property of $CsGeI_3$ was explored at the hydrostatic pressure from −0.5 GPa to 0.5 GPa. It has been found that its geometric structure, electronic structure, and elastic properties have changed significantly. When the applied pressure is −0.5 GPa, it has an optimal band gap value of 1.37 eV. Therefore, we focused on the comparative study of the electronic structure, optical properties, carrier migration properties, elastic properties, and thermodynamic properties when the pressure is −0.5 GPa and 0 GPa. Through the comparative analysis of these photoelectric properties, we found that $CsGeI_3$ after applied pressure is a good photoelectric material and suitable for PSC material.

## 2. Calculation Method

The first-principles based on DFT and plane wave pseudo-potential method implemented in the Cambridge Sequential Total Energy Package (CASTEP) code are used in the current calculation [26]. The interaction between electrons and ions is described by the OTFG Ultrasoft pseudo-potential, while the exchange correlation between electrons is described by Perdew–Burke–Ernzerhof (PBE) in generalized gradient approximation (GGA) [27]. We set the cut-off energy of the plane wave of the system at 435 eV, the monkhorst-pack scheme is used to calculate the Brillouin region, and the k grid point is set as $4 \times 4 \times 4$ [28], so that the energy and configuration of the system can converge at the level of quasi-complete plane wave base. The Pulay density mixing method is used in the self-consistent field operation; we set the self-consistent field as $5 \times 10^{-7}$ eV/atom. The Broyden–Fletcher–Goldfarb–Shanno (BFGS) algorithm is used to optimize the structure of the model. The convergence criteria of single atomic energy is set as $2 \times 10^{-5}$ eV, the convergence criteria of the interaction forces between atoms is set as 0.05 eV/Å, the convergence criteria of maximum displacement of atoms is set as 0.002 Å, and the convergence criteria of crystal internal stress is set as 0.1 GPa. CASTEP optimizes the four parameters at the same time and finally all meet the convergence criteria. The following valence electrons participate in the calculation: $Cs-5p^66s^1$, $Ge-4s^24p^2$, $Pb-6s^26p^2$, and $I-5s^25p^5$, respectively. Hydrostatic pressure changes from −0.5 GPa to 0.5 GPa in every 0.1 GPa.

As described in the literature, DFT-PBE significantly underestimates band gap calculation compared with experimental measurements. Perdew and Levy proved that the underestimation of band gaps in pure DFT is due in part to the lack of derivative continuity in the exchange correlation density functional, so the exchange correlation terms are undertreated [17]. Spin-orbit coupling GW

(soc-GW) quasiparticle corrections or spin-orbit coupling Heyd–Scuseria–Ernzerhof (soc-HSE) is an advanced calculation method with high precision and it has been proved to be effective in band gap calculations [29]. However, using DFT-PBE to discuss is still valid in this work. Studies have shown that compared to the band structure calculated by DFT-PBE, using soc-GW or soc-HSE to calculate neither changes the characteristics of band edge orbit nor the position of band gap in k-space [30]. Therefore, DFT-PBE can qualitatively provide an image of the evolution of band gap properties under pressure.

## 3. Results and Discussion

### 3.1. Crystal Structure

At room temperature, the crystal structure of perovskite $CsGeI_3$ belongs to the triangular crystal system and the space group is R-3m, as shown in Figure 1. The lattice constant of $CsGeI_3$ was found by Findit, which was geometrically optimized before calculation. The data are shown in Table 1. It can be seen from Table 1 that the optimized lattice constant and cell angle both increased to a certain extent compared with before, but the increased amplitude was very small. When the hydrostatic pressure of −0.5 GPa to 0.5 GPa is applied to $CsGeI_3$, the space group does not change.

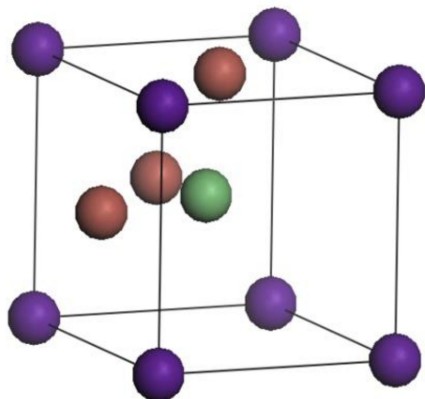

**Figure 1.** The crystal of $CsGeI_3$.

**Table 1.** Lattice parameters of $CsGeI_3$ with Findit compared with geometry optimization (GO).

|        | a = b = c (Å) | α = β = γ (°) | V (Å³) | Space Group |
|--------|---------------|---------------|--------|-------------|
| Findit | 5.98          | 88.61         | 213.98 | R-3m        |
| GO     | 6.08          | 88.76         | 224.55 | R-3m        |

As can be seen from Figure 2, the energy, lattice constant, volume, cell angle, bond length, and strain of the crystal are significantly changed after the pressure was applied. Figure 2a shows that the energy increases slightly after the pressure is applied, indicating that the stability of the crystal decreases. Figure 2b shows that the lattice length decreases (increases) as the pressure (tension) increases. Figure 2c shows that the volume decreases (increases) with the increase of pressure (tension). Figure 2d shows that the overall trend of cell angle increases from 87.5°to about 90°with the increase of pressure. Figure 2e shows the variation of the bond length of Ge-I with the pressure. It can be seen from Figure 2e that the bond length becomes shorter (becomes longer) with the increase of pressure (tension). Figure 2f shows the pressure-strain curve. With the increase of pressure (tension), the strain gradually decreases (increases). Where (b), (c), and (f) have similar changes, the lattice constant, volume, and strain satisfy the linear equation after the applied pressure as follows:

$$V = a^3, \ a = a_0(1 + \varepsilon) \tag{1}$$

where V is the crystal volume, a is the lattice constant after applied pressure, $a_0$ is the lattice constant without applied pressure, and $\varepsilon$ is the strain after applied pressure.

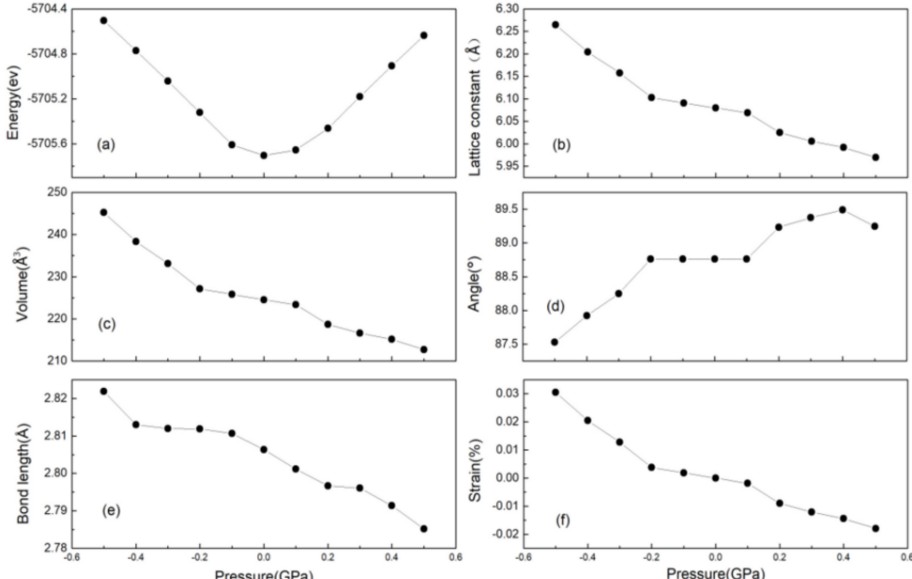

**Figure 2.** The structure parameters of CsGeI$_3$ under different pressure conditions. (**a**) The curve of energy. (**b**) The curve of the lattice constant. (**c**) The curve of volume. (**d**) The curve of cell angle. (**e**) The curve of bond length of Ge-I. (**f**) The curve of the strain.

The structural parameters of the crystal changed obviously after applied pressure, which will affect the electronic structure of the crystal.

### 3.2. Electronic Structure

In order to study the effect of pressure on the electronic structure, we studied the effect of band structure and density of states (DOS) under pressure. We obtained the band gap value of 0.97 eV when no pressure was applied, which is similar to most theoretical calculations [31], but much smaller than the experimental result of 1.60 eV [32]. Although the theoretical calculation is smaller than the experimental value, it does not affect our study on its properties. Therefore, we continued to use DFT-PBE to study the changes of photoelectric properties of perovskite CsGeI$_3$ after applied pressure. We applied pressure from −0.5 GPa to 0.5 GPa to the material and found that the band gap gradually decreased from 1.37 eV to 0.72 eV, as shown in Figure 3. It was found that when the material is applied with pressure, the band gap gradually decreases, while when the material is applied with tension, the band gap gradually increases. When the tension is 0.5 GPa, the band gap value is 1.37 eV, between 1.3–1.4 eV. According to the Shockley–Queisser theory, this is the band gap value with the best photoelectric efficiency of perovskite materials [33]. As can be seen from Figure 3, the band gap value changed significantly after applied pressure, which is due to the change in the electronic structure of the material. It can be seen from Figure 2 that the cell angle of CsGeI$_3$ and the bond length of Ge-I all changed significantly.

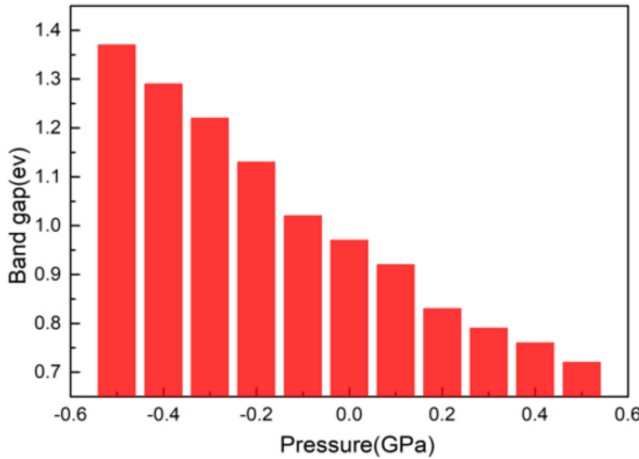

**Figure 3.** The band gap of CsGeI$_3$ under different pressure conditions.

We focused on the comparative study of band structure and DOS when the pressure is −0.5 GPa and 0 GPa. As shown in Figures 4 and 5, the band structure and DOS barely changed after the pressure was applied. It can be seen from the figure of band structure that CsGeI$_3$ is still a direct band gap semiconductor, and the conduction band becomes a little flatter after applied pressure. From Figure 5, it can be seen that the Ge-4p orbital mainly contributes to the conduction band minimum (CBM), and valence band maximum (VBM) was mainly contributed by I-5p orbital. The Cs atom mainly plays a role in the 5p electron orbital, but its DOS is mainly in the deep energy level above 6 eV and below −6 eV, and its contribution to the Fermi surface is almost zero. However, Cs atom is very important to stabilize the structure of CsGeI$_3$.

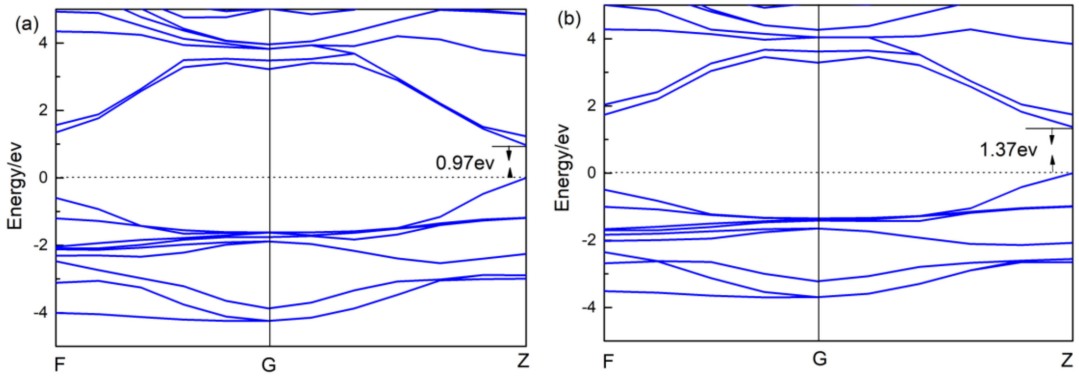

**Figure 4.** The band structure of CsGeI$_3$ under the pressure of (**a**) 0 GPa and (**b**) −0.5 GPa.

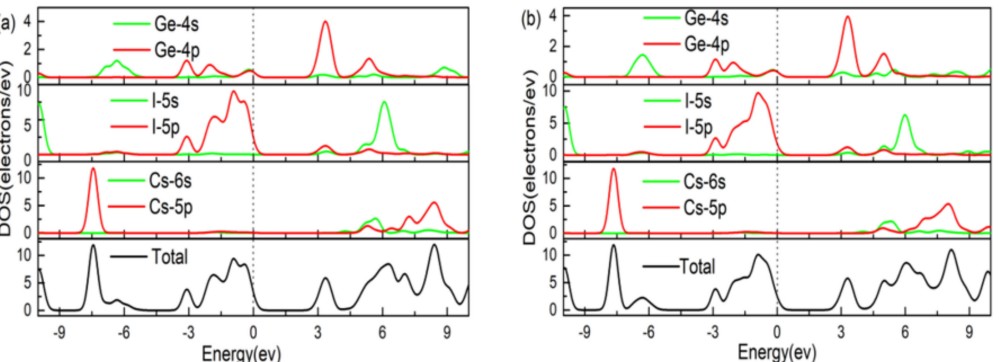

**Figure 5.** The DOS of CsGeI$_3$ under the pressure of (**a**) 0 GPa and (**b**) −0.5 GPa.

To further understand the electrons under pressure, we also studied the charge transfer in CsGeI$_3$, as shown in Figure 6. When the pressure changed from −0.5 GPa to 0.5 GPa, the charges of Cs and Ge gradually decrease from 0.61 e and 0.22 e to 0.56 e and 0.08 e, respectively. According to the law of conservation of charge, the charge of I also decreased. The calculation shows that the charge of I decreased from −0.28 e to −0.21 e.

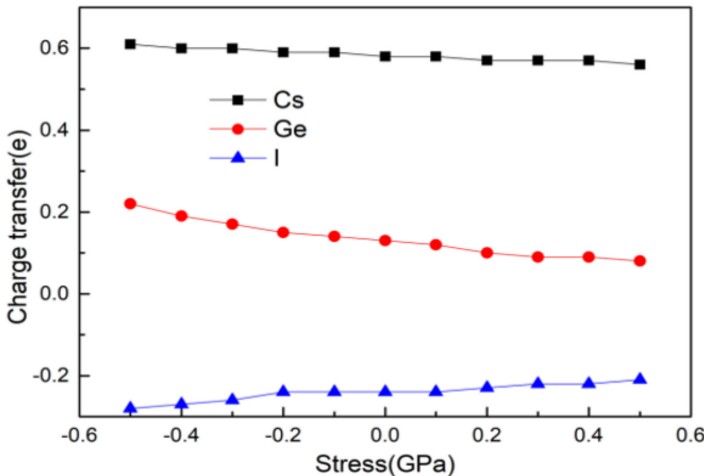

**Figure 6.** The charge transfer of the Cs, Ge, and I atoms.

*3.3. Optical Properties*

Perovskite CsGeI$_3$ can be widely used in solar cells materials, so it is particularly important to study its optical properties. The optical properties of solids can be described by the dielectric function $\varepsilon(\omega) = \varepsilon_1(\omega) + i\varepsilon_2(\omega)$. The dielectric function consists of intraband and interband transitions. Intraband transitions mainly occur in metals and interband transitions mainly occur in semiconductors. Interband transitions consist of direct and indirect transitions. Since the indirect transition only involves phonon dispersions and has little influence on the dielectric function, it can be ignored. The contribution of the direct interband transitions to the imaginary dielectric function $\varepsilon_2(\omega)$ can be obtained from the Kohn–Sham particle equation [1]:

$$\varepsilon_2(\omega) = \frac{Ve^2}{2\pi\hbar m^2\omega^2}\int d^3k\sum_{nn'}\left|\langle kn|p|kn\rangle'\right|^2 f(kn)\times(1-f(kn'))\delta(E_{kn}-E_{kn'}-\hbar\omega) \tag{2}$$

where $V$ is the unit volume, $e$ is the electron charge, $\hbar$ is the reduced Planck Constant, and $p$ is the momentum transition matrix. The wave functions of VB and VB are expressed in terms of $kn$ and $kn'$. The real part can be obtained through the Kramers–Kronig relation [34], where M is the principal component value of the integral.

$$\varepsilon_1 = 1 + \frac{2}{\pi}M\int_0^\infty \frac{\varepsilon_2(\omega')\omega'}{\omega'^2-\omega^2}d\omega \tag{3}$$

The real part of the dielectric constant is used to describe the propagation behavior of the electromagnetic field in the material, and the imaginary part of the dielectric constant is used to describe absorption of light in the material. Figure 7 shows the real and imaginary parts of the dielectric function of CsGeI$_3$ when the energy range is 0 to 10 eV and the pressure is 0 GPa and −0.5 GPa. It can be seen from Figure 7 that the curves of their dielectric functions are very similar. As evident in Figure 7a, the static dielectric constants of pressurized and unpressurized are 5.5 and 5.2, respectively. $\varepsilon_1$ reached its peak value near 3 eV, the peak value of unpressurized was 6.3, and the peak value of pressurized was 6.7. Although there are peaks in the future, the size of the peak gradually decreases.

It is worth noting that $\varepsilon_1$ is negative around 9 eV. A negative value indicates that light cannot be transmitted to the material and that the material exhibits a certain metallic luster. It can be seen from Figure 7b that the thresholds of $\varepsilon_2$ starts from about 1 eV and 1.4 eV respectively. These thresholds are called basic absorption edges [35] and are related to the direct interband transition between VBM and CBM in CsGeI$_3$, which is consistent with the band values we calculated previously. $\varepsilon_2$ reaches the first peak at about 4.0 eV, the second peak (maximum) at about 6.5 eV, and the third peak at about 8.5 eV. The first peak value is 3.7 and 4.1, the second peak value is 4.3 and 4.9, and the third peak value is 3.4 and 3.6 when the applied pressure is 0 GPa and −0.5 GPa. Combined with the DOS in Figure 5, it can be seen that the first peak is mainly caused by the transition of electrons in Ge atom and I atom, the second peak is mainly caused by the transition of electrons in Cs atom and I atom, and the third peak is determined by the complex transition of electrons in the deep energy levels of Cs, Ge, and I. By comparison, we find that the corresponding peak value when the pressure is −0.5 GPa is higher than the peak value when the pressure is 0 GPa. This indicates that CsGeI$_3$ has better dielectric properties after applied pressure.

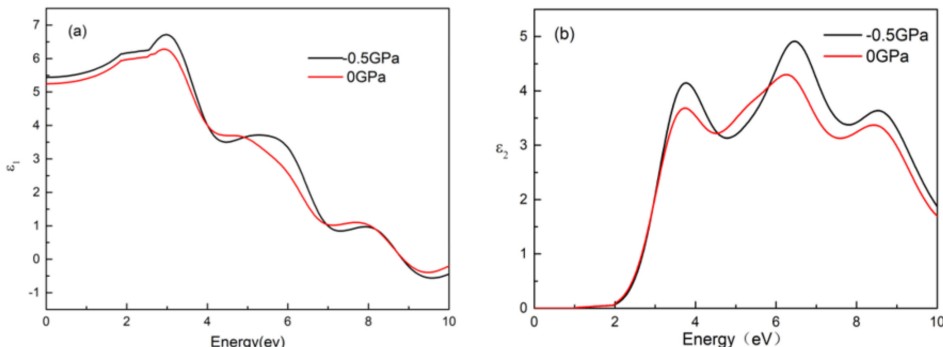

**Figure 7.** The dielectric function of CsGeI$_3$ of (**a**) real and. (**b**) imaginary under the pressure of 0 GPa and −0.5 GPa.

Figure 8 shows the real and imaginary parts of the conductivity of the perovskite CsGeI$_3$. We know that the real part of the conductivity represents the actual dissipated energy, and that when the moving electrons hit the lattice and get dragged down, they transfer their kinetic energy to the lattice and eventually they dissipate as joule heat. The imaginary part of the conductivity represents the conversion of the energy of the electric field to the kinetic energy of the electrons, which makes the electrons do simple harmonic vibrations under the action of an external alternating electric field. We can see from Figure 8a that the real part of the conductivity is similar to the imaginary part of the dielectric function. The value of the threshold and the position of the peak value coincide with the imaginary part of the dielectric function. At 6.5 eV, the real part of the conductivity has a second peak (maximum) of 3.3 (0 GPa) and 3.9 (−0.5 GPa), respectively. Figure 8b shows that the imaginary part of the conductivity of them is not significantly changed. The peak value of the real part of the conductivity is slightly larger after applied pressure, indicating that the energy loss is larger.

Figure 9 shows the absorption coefficient, from which it is obvious that there are three absorption peaks, which appear in positions consistent with the imaginary part of the dielectric function and the real part of the conductivity. It can be seen that the dielectric function, conductivity, and absorption coefficient are closely related. The absorption coefficient can be described by the real and imaginary parts of the dielectric function as follows [36]:

$$I(\omega) = 2\omega \left( \frac{\left[\varepsilon_1^2(\omega) + \varepsilon_2^2(\omega)\right]^{1/2} - \varepsilon_1(\omega)}{2} \right)^{1/2} \tag{4}$$

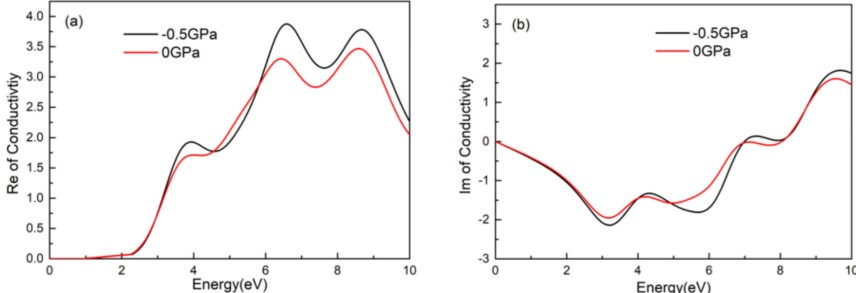

**Figure 8.** The conductivity of CsGeI$_3$ of (**a**) real and (**b**) imaginary under the pressure of 0 GPa and −0.5 GPa.

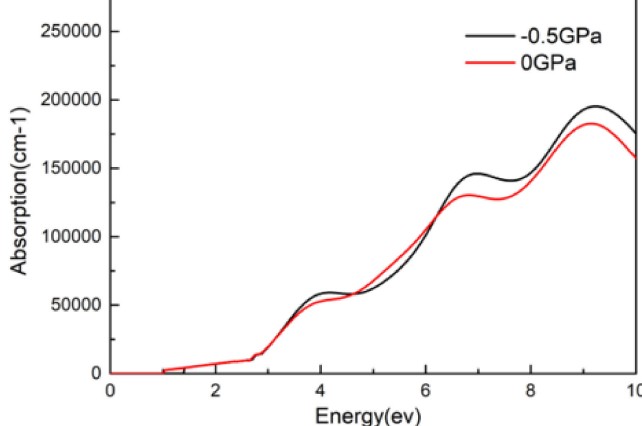

**Figure 9.** The absorption of CsGeI$_3$ under the pressure of 0 GPa and −0.5 GPa.

It can be seen from Figure 9 that the absorption peak under pressure is slightly larger than that without pressure. Be it in the visible or the ultraviolet region, the absorption effect after pressure is better, which indicates that the photoelectric performance after pressure is better and more suitable for solar cells materials. Moreover, we found that the absorbed light has a blue shift after pressure, indicating that the absorbed energy increases, which is consistent with the increase of band gap after pressure. Perovskite CsGeI$_3$ is highly absorptive in a wide energy range near the main peak, so it is suitable for use as the absorption layer of solar cells.

*3.4. Carrier Transport Properties*

After absorbing the photons, the electrons transition from the VB to the CB, resulting in electrons and holes. We can better understand their PCE by calculating their effective masses and exciton binding energy. The calculation of effective mass satisfies the following formula:

$$\frac{1}{m^*} = \frac{1}{\hbar^2} \frac{\partial^2 E_n(\vec{k})}{\partial k^2} \tag{5}$$

As shown in Figure 10, we selected the curves of conduction band minimum and valence band maximum, combined Equation (5) and the software of origin, and obtained the effective masses of electrons and holes at the lowest point (G point) of the band gap; the calculated results are summarized in Table 2. According to Table 2, the electron effective mass is 0.27 $m_0$ and the hole effective mass is 0.34 $m_0$ at 0 GPa; the electron effective mass is 0.23 $m_0$ and the hole effective mass is 0.25 $m_0$ at −0.5 GPa. It can be seen that the effective mass of electrons and holes decreases correspondingly after applied pressure, which will lead to better migration efficiency and improve the PCE of solar cells. When looking into the literature, we did not find the experimental data, so we cannot compare with the experimental data, but our calculation provides a reference for future experiments.

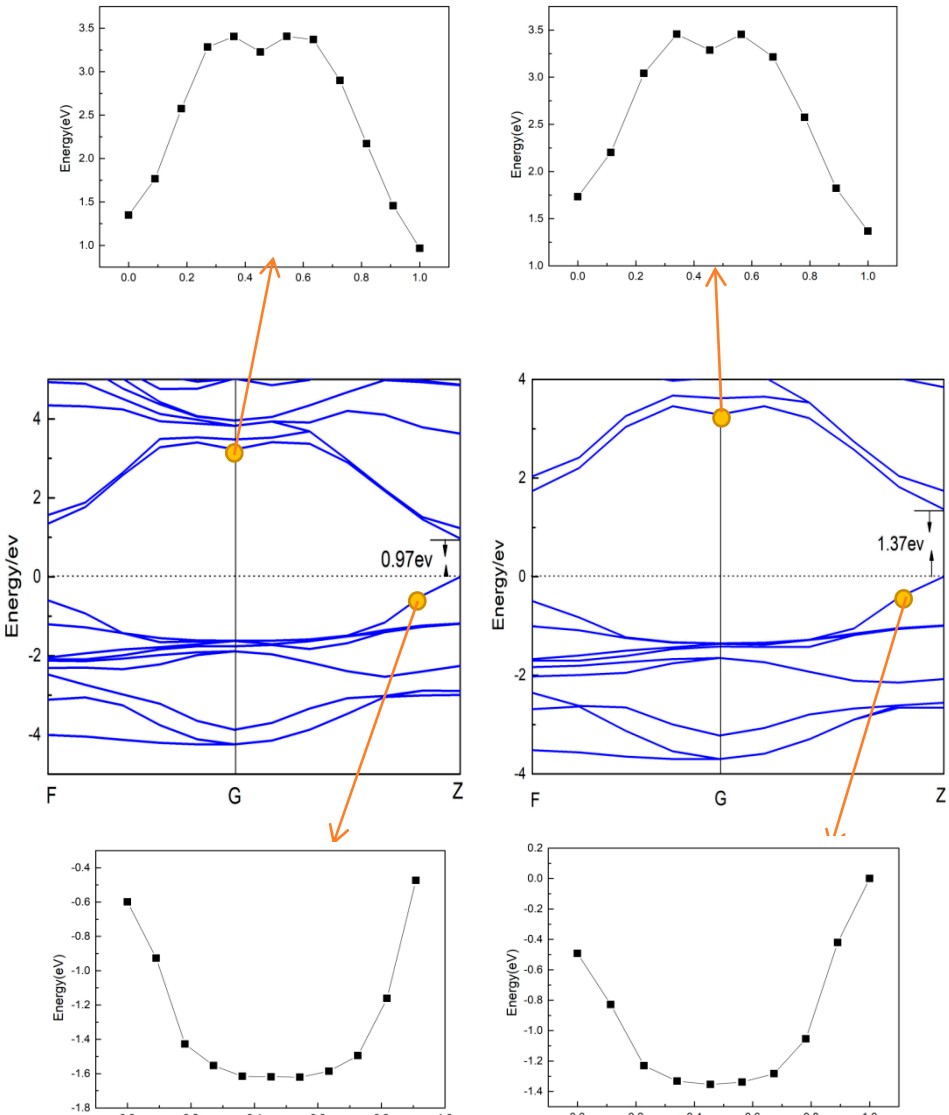

**Figure 10.** Calculated of effective mass at Z point of conduction band bottom and valence band top of CsGeI$_3$ under the pressure of 0 GPa and −0.5 GPa.

**Table 2.** Effective masses and exciton binding energy calculated for CsGeI$_3$ under the pressure of 0 GPa and −0.5 GPa. Masses are given in units of the free electron mass m$_0$.

|          | m$_e$ | m$_h$ | μ    | ε$_s$ | E$_b$ (ev) |
|----------|-------|-------|------|-------|-----------|
| 0 GPa    | 0.27  | 0.34  | 0.15 | 5.2   | 0.075     |
| −0.5 GPa | 0.23  | 0.25  | 0.12 | 5.5   | 0.054     |

The exciton binding energy was calculated within the weak Mott–Wannier model, using the following formula [37]:

$$E_b = 13.56 \frac{\mu}{\varepsilon_s^2} \tag{6}$$

where $\mu$ is the effective reduced mass, which satisfies the formula:

$$\frac{1}{\mu} = \frac{1}{m_e^*} + \frac{1}{m_h^*} \tag{7}$$

where $\varepsilon_s$ is the static dielectric constant (as previously seen). It can be seen from Table 2 that the exciton binding energy is 0.075 eV without pressure, and 0.054 eV under applied pressure. The lower exciton binding energy means that photogenic carriers (electrons and holes) are more easily generated, which helps to improve photoelectric conversion efficiency.

By calculating the effective mass and the exciton binding energy, we found that $CsGeI_3$ has a lower effective mass and lower exciton binding energy and can be used as efficient light absorbing material.

### 3.5. Elastic Properties

For a mechanically stable triangular structure, these elastic constants in Table 3 must satisfy the Born–Huang stability criteria given by [38]:

$$C_{11} - |C_{12}| > 0, (C_{11} + C_{12})C_{33} > 2C_{13}^2, (C_{11} - C_{12})C_{44} > 2C_{14}^2 \tag{8}$$

**Table 3.** Calculated elastic constant, Bulk modulus (B), Shear modulus (G), and elastic anisotropy (A) of $CsGeI_3$ under the pressure of 0 GPa and −0.5 GPa.

|  | $C_{11}$ | $C_{12}$ | $C_{13}$ | $C_{14}$ | $C_{33}$ | $C_{44}$ | B | G | A |
|---|---|---|---|---|---|---|---|---|---|
| 0 GPa | 60.07 | 48.61 | 32.61 | 22.32 | 74.74 | 23.03 | 46.95 | 10.54 | 4.02 |
| −0.5 GPa | 48.52 | 38.88 | 25.83 | 17.21 | 61.03 | 13.64 | 37.68 | 15.11 | 2.83 |

By substituting the elastic constants in Table 3 into the above criteria conditions, it can be seen that the calculated results meet the stability conditions under the hydrostatic pressures of 0 GPa and −0.5 GPa, which indicates that the crystal structure of $CsGeI_3$ is stable.

The stability tolerance factor of $ABX_3$ can be obtained by Goldschmidt criterion [39]:

$$T = \frac{R_A + R_X}{\sqrt{2}(R_B + R_X)} \tag{9}$$

For organic-inorganic $ABX_3$ halides, Li et al. found that the stability conditions should meet $0.81 < T < 1.11$ [40]. According to the calculated results in Table 4, the crystal structure of $CsGeI_3$ is stable, which is consistent with the calculated results above.

**Table 4.** The ionic radium of $CsGeI_3$.

|  | $Cs^+$ | $Ge^{2+}$ | $I^-$ | T |
|---|---|---|---|---|
| R (nm) | 0.167 | 0.073 | 0.22 | 0.93 |

Figure 11 shows phonon spectra of $CsGeI_3$ under both pressure conditions. It can be seen from the figure that there is no virtual frequency, which further indicates that their structure is stable.

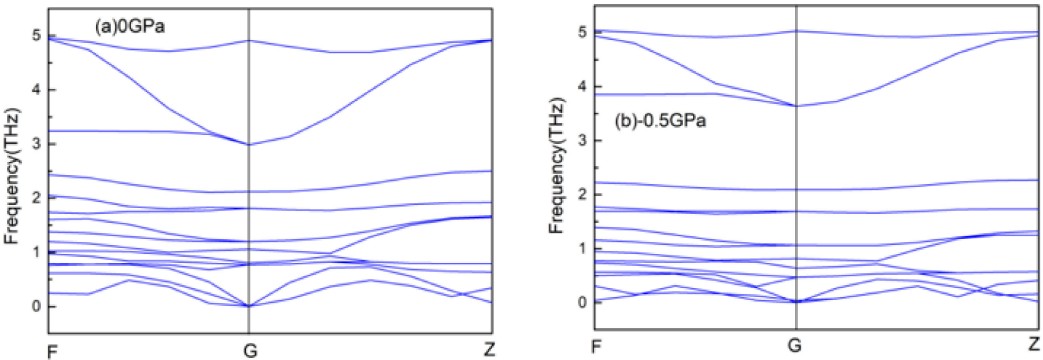

**Figure 11.** Phonon spectrum of $CsGeI_3$ under the pressure of (**a**) 0 GPa and (**b**) −0.5 GPa.

The Bulk modulus is a measure to determine the strength or hardness of a crystal and it represents the brittleness resistance of a crystal, so the greater the Bulk modulus, the greater the strength of the crystal. As can be seen from Table 3, the Bulk modulus of perovskite CsGeI$_3$ is 46.95 GPa when there is no hydrostatic pressure, and 37.68 GPa when the hydrostatic pressure is −0.5 GPa. To understand the strength of crystals, this value can be compared with the Bulk modulus of glass (35–50 GPa) and diamond (443 GPa), indicating that the perovskite CsGeI$_3$ is a soft material under both pressure conditions. We found that the Bulk modulus decreased after applied the hydrostatic pressure of −0.5 GPa. This is due to the application of tension, resulting in a decrease of the density and hardness of the material. Shear modulus represents the ability of a material to resist shear strain. It can be seen from Table 3 that the shear modulus increases after the tension is applied, indicating that its transverse resistance to strain is enhanced. Based on the elasticity coefficient, Pugh proposed a standard to judge the ductility or brittleness of materials [41]: when B/G > 1.75, the material is ductile, otherwise it is brittle. It can be seen from Table 3 that the corresponding ratios of 0 GPa and −0.5 GPa are 4.45 and 2.49, both of which are greater than 1.75. As such, this compound has the properties of softness and ductility, and can be used for automobiles and curved surfaces of glasses.

In order to judge material anisotropy, Ranganathan and Ostoja-Starzewski introduced the criteria for judging material anisotropy [42]:

$$A = \frac{2C_{44}}{C_{11} - C_{12}} \tag{10}$$

$A$ is the elastic anisotropy coefficient. If $A = 1$, the material is isotropic, otherwise the material is anisotropic. According to Table 2, when the hydrostatic pressure is 0 GPa, $A = 4.02$; and when the hydrostatic pressure is −0.5 GPa, $A = 2.83$. Therefore, the perovskite CsGeI$_3$ is anisotropic in both cases.

### 3.6. Thermodynamic Properties

We calculated the properties of phonons to find the thermodynamic properties of Debye temperature, heat capacity, entropy, enthalpy, and free energy that vary with temperature. It is worth noting that for all the thermodynamic temperature ranges we chose 0–1000 K. Figure 12a shows the change of Debye temperature with thermodynamic temperature under the pressure of 0 GPa and −0.5 GPa. We found that the Debye temperature increased with the increase of thermodynamic temperature, but it increased significantly faster under the pressure of −0.5 GPa, indicating that the thermal conductivity of CsGeI$_3$ was higher after pressure. However, within the thermodynamic temperature of 1000 K, neither of them reached the maximum Debye temperature. Figure 12b shows the change of heat capacity with thermodynamic temperature. The heat capacity and Debye temperature and the thermodynamic temperature satisfy the following relation:

$$C_V = \frac{12\pi^4}{5} R \left( \frac{T}{\theta_D} \right)^3 \tag{11}$$

where $R$ is Rydberg constant and $\theta_D$ is Debye temperature. From Figure 12b, we can see that the heat capacity increases with the increase of temperature and finally tends to be stable. The maximum heat capacity corresponding to 0 GPa and −0.5 GPa are 29.6 J/cell.k and 29.1 J/cell.k, so the heat capacity of 0 GPa is slightly higher. Combined with Equation (11) and Figure 12a, the Debye temperature of 0 GPa is lower at the same temperature, so the heat capacity is higher, which is consistent with the results obtained in Figure 12b. Finally, their heat capacity approaches a constant value ($C_V = 3R$), independent of temperature, satisfying Dulong-Petit's Law.

Figure 12c shows the relationship between Gibbs free energy, enthalpy, and entropy with temperature under both pressure conditions. Gibbs free energy, enthalpy, and entropy satisfy the following relationship:

$$\Delta G = \Delta H - T\Delta S \tag{12}$$

It can be seen from Figure 12c that entropy and Gibbs free energy have great influence after pressure, while enthalpy has little influence. As the temperature rises, the thermal motion and lattice vibration

of particles will increase, and the degree of chaos in the system will increase, eventually leading to the increase of entropy. The change of enthalpy with temperature is less than TS, which leads to the decrease of free energy with the increase of temperature. The variation trend of the three is consistent with Equation (12). The Gibbs free energy without pressure decreases faster with the increase of temperature, indicating that Gibbs free energy has higher stability without pressure.

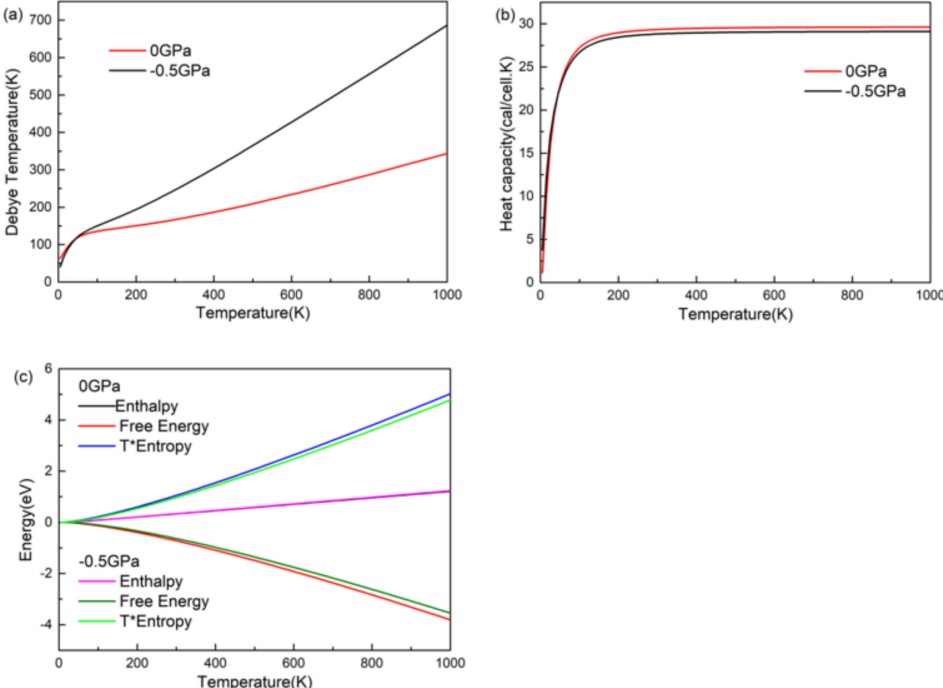

**Figure 12.** The thermodynamic properties of CsGeI$_3$ of (**a**) Debye temperature, (**b**) heat capacity, and (**c**) enthalpy, temperature-entropy, and free energy under the pressure of 0 GPa and −0.5 GPa.

## 4. Conclusions

Based on the first-principles, the physical properties of CsGeI$_3$ were studied under hydrostatic pressure from −0.5 GPa to 0.5 GPa with an interval of 0.1 GPa in this paper. The results showed that its geometric structure, electronic structure, and elastic properties changed significantly. When the hydrostatic pressure was −0.5 GPa, the optical band gap value of CsGeI$_3$ increased from 0.97 eV to 1.37 eV, reaching the optimal band gap value. This paper focused on the properties of CsGeI$_3$ when the pressure was 0 GPa and −0.5 GPa, and the results are as follows:

1.  CsGeI$_3$ is still a direct band gap semiconductor. The band structure and DOS of CsGeI$_3$ hardly changed after applied pressure, but the conduction band became a little flatter. When the pressure changed from −0.5 GPa to 0.5 GPa, the charge transfer number of Cs$^+$, Ge$^{2+}$, and I$^-$ decreased.
2.  The dielectric, conductivity, and absorption coefficient of CsGeI$_3$ under pressure of −0.5 GPa were higher than that without pressure. Three peaks appeared simultaneously in the imaginary part of the dielectric function, the real part of the conductivity, and the absorption coefficient, located around 4.0 eV, 6.5 eV, and 8.5 eV. In either visible light or an ultraviolet region, the absorption peak was slightly larger after applied pressure than that without applied pressure, and the absorption light had a blue shift phenomenon under pressure.
3.  Both the effective mass and exciton binding energy were reduced after the application of pressure, indicating that photogenic carriers were more likely to be generated after pressure, and that carriers had better migration efficiency, which was conducive to improving the photoelectric conversion efficiency.

4. Though multiple calculations of the Born–Huang stability criterion, the tolerance factor T, and phonon spectrum with or without virtual frequency, it was found that $CsGeI_3$ was stable under both pressure conditions. We also calculated the elastic modulus of both pressure conditions and found that they were both soft, ductile, and anisotropic, and the values of B, B/G, and A decreased after applying pressure.

5. It was found that the Debye temperature and heat capacity increased with the increase of thermodynamic temperature, and the Debye temperature increased rapidly after pressure, while the heat capacity increased slowly and eventually stabilized. Through the calculation of enthalpy, entropy, and Gibbs free energy of $CsGeI_3$, it was found that the Gibbs free energy of $CsGeI_3$ decreases faster with the increase of temperature without pressure, which indicates that $CsGeI_3$ has higher stability without pressure.

**Author Contributions:** Conceptualization, Formal analysis, Methodology, and Writing—Original draft, L.-K.G.; Funding acquisition, Supervision, Y.-L.T.; Writing—Review and editing, Y.-L.T., X.-F.D. All authors have read and agreed to the published version of the manuscript.

**Funding:** This work is supported by the National Natural Science Foundation of China Program (Grant No.11164004, 61835003), Guizhou Provincial Photonic Science and Technology Innovation team (Qianke Joint talents team [2015]4017) and the Industrial Research Project of Guizhou Province (GY[2012]3060).

**Conflicts of Interest:** The authors declare no conflict of interest.

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
