# Peer review of "First-Principles Study on the Photoelectric Properties of CsGeI3 under Hydrostatic Pressure"

_applsci, doi:10.3390/app10155055_

Round 1

Reviewer 1 Report

The Authors present reasonable study for interesting idea to suport material conductivity.

Author Response

Thank you very much for your review to my manuscript.

Reviewer 2 Report

The photoelectric properties of CsGeI3 by applying successive hydrostatic pressure were studied by the authors. CsGeI3 has higher dielectric value, conductivity and absorption coefficient and blue shift in absorption spectrum when the pressure is ‐0.5 GPa. CsGeI3 is an efficient light absorbing material, as show the experimental calculating results of the effective mass of electron and hole and the exciton binding energy. Also, CsGeI3 is stable under both pressure conditions through multiple calculations of the Born Huang stability criterion, tolerance factor T and phonon spectrum with or without virtual frequency. The Gibbs free energy decreases faster with the increase of temperature without applied pressure, which indicates that CsGeI3 has higher stability without pressure.

The paper deserves to be published after its revision.

-methylamine lead iodine (perovskite materials) should be methylammonium lead iodide (R 37, 52, 56)
-methylether lead iodine should be methylether lead iodide (R 38, 39)
- I think it should be mentioned that methylammonium cation is MA+ and formamidinium cation is FA+. (R 38, 41)
- I think the paper "Pressure-induced effects in the inorganic halide perovskite CsGeI3", by Diwen Liu,   Qiaohong Li,   Huijuan Jing and  Kechen Wu, from RSC Advances 2019, should be cited, because the subject is very similar.
- Please verify Figure 1. The crystal of CsGeI3.
- Please check equation (2), equation (3)and equation (4).

-Please write the bibliographic references correctly

Author Response

The photoelectric properties of CsGeI3 by applying successive hydrostatic pressure were studied by the authors. CsGeI3 has higher dielectric value, conductivity and absorption coefficient and blue shift in absorption spectrum when the pressure is ‐0.5 GPa. CsGeI3 is an efficient light absorbing material, as show the experimental calculating results of the effective mass of electron and hole and the exciton binding energy. Also, CsGeI3 is stable under both pressure conditions through multiple calculations of the Born Huang stability criterion, tolerance factor T and phonon spectrum with or without virtual frequency. The Gibbs free energy decreases faster with the increase of temperature without applied pressure, which indicates that CsGeI3 has higher stability without pressure.

The paper deserves to be published after its revision.

-methylamine lead iodine (perovskite materials) should be methylammonium lead iodide (R 37, 52, 56)

-methylether lead iodine should be methylether lead iodide (R 38, 39)

- I think it should be mentioned that methylammonium cation is MA+ and formamidinium cation is FA+. (R 38, 41)

We’ve revised them in page 1 lines 37-40.

- I think the paper "Pressure-induced effects in the inorganic halide perovskite CsGeI3

", by Diwen Liu, Qiaohong Li, Huijuan Jing and Kechen Wu, from RSC Advances 2019, should be cited, because the subject is very similar.

Although Liu et al. studied some photoelectric properties of CsGeI3 under hydrostatic pressure through first-principles, we still do not know enough about it(page 2,lines 68-69).

 - Please verify Figure 1. The crystal of CsGeI3.

We’ve verified Figure 1.

- Please check equation (2), equation (3)and equation (4).

We’ve checked them.

-Please write the bibliographic references correctly

We’ve checked them.

Reviewer 3 Report

Manuscript #applsci-861271
======================

Manuscript entitled "First-principles study on the photoelectric properties
of CsGeI3 under hydrostatic pressure" considers the properties of the
novel photoactive inorganic material under various pressure conditions. The
problem to be solved is of great importance for the design of new materials
for practical applications in solar cells or photovoltaic devices. However
the poor English seems to be a crucial problem of the manuscript - it is
hard to read and to understand. Moreover, it is written in a hermetic way,
difficult to understand for people not working with inorganic perovskites.

My objections also considers the lack of the practical aspects of pressure
applied explained in the introduction. The manuscript treats the
hydrostatic pressure of negative values, which is not obvious for general
chemists or physicists and needs to be commented upon.

Trivial statements such as "Thermodynamics is the study of some physical properties
of materials at different temperatures, which is very important for studying the influence of temperature on materials." must be avoided in high-impact journals (page 11, line 303).

It needs a revision, mostly with respect to the language applied, to
be improved. Some exemplary unclear sentences are given below:
* page 5, line 158: According to the law of conservation of charge, I has the same law
* page 6, line 183: Although there are peaks in the future, but the overall trend is gradually reduced.
* page 6, line 185: We know that a negative value means that light cannot travel
to the perovskite material and that the material exhibits metallic luster.
* page 8, line 237: we select the curves of conduction band minimum and valence band
maximum, combine formula (5) and origin mapping software.
* page 9, line 259: By calculating the effective mass and the exciton binding energy, we find that they have lower
effective mass and lower exciton binding energy - who is 'they'?
* page 10, line 265: For a mechanically stable triangular structure, these elastic constants must satisfy the Born
Huang stability criteria - what means 'these' in the first sentence of the
subsection?
* page 10, line 275: For organic‐inorganic ABX 3 halides, Li et al. showed that stable perovskite needs to meet
0.81<T<1.11
* page 11, line 327: entropy and Gibbs free energy are mainly affected after pressure,
while enthalpy has little influence
* page 11, line 329: the thermal motion and lattice vibrations
of the particles increase, thus increasing the degree of confusion, leading to an increase in entropy

Such a language rather leads to the confusion of a reader than increase of
entropy.

Misprints and errors also need to be corrected, such as:
* page 4, line 139: According to Shockley‐.
* page 5, lines 146, 147: that the 4P of Ge mainly contributes, was mainly
contributed by 5P of I - if P is to be denote electron shells, it should
be given in small letters and explicitely written

In general, after the careful language correction and improvement of the
above mentioned issues, I do recommend this manuscript for a
re-consideration.

Author Response

Comments and Suggestions for Authors

Manuscript entitled "First-principles study on the photoelectric properties of CsGeI3 under hydrostatic pressure" considers the properties of the novel photoactive inorganic material under various pressure conditions. The problem to be solved is of great importance for the design of new materials for practical applications in solar cells or photovoltaic devices. However the poor English seems to be a crucial problem of the manuscript - it is hard to read and to understand. Moreover, it is written in a hermetic way,difficult to understand for people not working with inorganic perovskites.

My objections also considers the lack of the practical aspects of pressure applied explained in the introduction. The manuscript treats the hydrostatic pressure of negative values, which is not obvious for general chemists or physicists and needs to be commented upon.

Trivial statements such as "Thermodynamics is the study of some physical properties of materials at different temperatures, which is very important for studying the influence of temperature on materials." must be avoided in high-impact journals (page 11, line 303).

We've removed it(page 14, lines 369-370).

It needs a revision, mostly with respect to the language applied, to be improved. Some exemplary unclear sentences are given below:* page 5, line 158: According to the law of conservation of charge, I has the same law

According to the law of conservation of charge, the charge of I also decrease. The calculation shows that the charge of I decreases from -0.28e to -0.21e(page 7, lines 193-194).
* page 6, line 183: Although there are peaks in the future, but the overall trend is gradually reduced.

Although there are peaks in the future, the size of the peak is gradually decreasing(page 8, lines 229-230).
* page 6, line 185: We know that a negative value means that light cannot travel to the perovskite material and that the material exhibits metallic luster.

A negative value indicates that light cannot be transmitted to the material and that the material exhibits a certain metallic luster(page 8, lines 232-233).
* page 8, line 237: we select the curves of conduction band minimum and valence band maximum, combine formula (5) and origin mapping software.

we select the curves of conduction band minimum and valence band maximum, combine formula (5) and software of origin, and obtain the effective masses of electrons and holes at the lowest point(G point) of the band gap, and summarize the calculated results in Table 2(page 11, lines 298-300).
* page 9, line 259: By calculating the effective mass and the exciton binding energy, we find that they have lower effective mass and lower exciton binding energy - who is 'they'?

By calculating the effective mass and the exciton binding energy, we find that CsGeI3 has lower effective mass and lower exciton binding energy(page 12, line 321)
* page 10, line 265: For a mechanically stable triangular structure, these elastic constants must satisfy the Born Huang stability criteria - what means 'these' in the first sentence of the subsection?

For a mechanically stable triangular structure, these elastic constants in Table 3 must satisfy the Born Huang stability criteria(page 13, line 327)
* page 10, line 275: For organic‐inorganic ABX 3 halides, Li et al. showed that stable perovskite needs to meet 0.81<T<1.11

For organic-inorganic ABX3 halides, Li et al. found that the stability conditions should meet 0.81<T<1.11(page 13, lines 337-339).
* page 11, line 327: entropy and Gibbs free energy are mainly affected after pressure,while enthalpy has little influence

It can be seen from Figure 12(c) that entropy and Gibbs free energy have great influence after pressure, while enthalpy has little influence(page 14, lines 394-395).
* page 11, line 329: the thermal motion and lattice vibrations of the particles increase, thus increasing the degree of confusion, leading to an increase in entropy

Such a language rather leads to the confusion of a reader than increase of entropy.

As the temperature rises, the thermal motion and lattice vibration of particles will increase, and the degree of chaos in the system will increase, eventually leading to the increase of entropy(page 14-15, lines 395-397).

Misprints and errors also need to be corrected, such as:* page 4, line 139: According to Shockley‐.

Due to the editor lost the original content in typesetting, we are now making it up in page 5 lines 168-172.
* page 5, lines 146, 147: that the 4P of Ge mainly contributes, was mainly contributed by 5P of I - if P is to be denote electron shells, it should be given in small letters and explicitely written

From Figure 5, it can be seen that the Ge-4p orbital mainly contributes to the conduction band minimum(CBM), and valence band maximum (VBM) was mainly contributed by I-5p orbital. Cs atom mainly plays a role in the 5p electron orbital(page 6, lines 180-182)

In general, after the careful language correction and improvement of the above mentioned issues, I do recommend this manuscript for a re-consideration.
